# Design, Synthesis and Antifungal/Nematicidal Activity of Novel 1,2,4-Oxadiazole Derivatives Containing Amide Fragments

**DOI:** 10.3390/ijms23031596

**Published:** 2022-01-29

**Authors:** Dan Liu, Ling Luo, Zhengxing Wang, Xiaoyun Ma, Xiuhai Gan

**Affiliations:** 1State Key Laboratory Breeding Base of Green Pesticide and Agricultural Bioengineering/Key Laboratory of Green Pesticide and Agricultural Bioengineering, Ministry of Education, Guizhou University, Huaxi District, Guiyang 550025, China; liudan19960516@163.com (D.L.); ll17792015@163.com (L.L.); wzx2669452150@163.com (Z.W.); 2School of Chemistry and Materials Science, Guizhou Education University, Wudang District, Guiyang 550018, China; maxiaoyun821215@163.com

**Keywords:** 1,2,4-oxadiazole, antifungal activity, nematicidal activity, succinate dehydrogenase

## Abstract

Plant diseases that are caused by fungi and nematodes have become increasingly serious in recent years. However, there are few pesticide chemicals that can be used for the joint control of fungi and nematodes on the market. To solve this problem, a series of novel 1,2,4-oxadiazole derivatives containing amide fragments were designed and synthesized. Additionally, the bioassays revealed that the compound **F15** demonstrated excellent antifungal activity against *Sclerotinia sclerotiorum* (*S. sclerotiorum*) in vitro, and the EC_50_ value of that was 2.9 μg/mL, which is comparable with commonly used fungicides thifluzamide and fluopyram. Meanwhile, **F15** demonstrated excellent curative and protective activity against *S. sclerotiorum*-infected cole in vivo. The scanning electron microscopy results showed that the hyphae of *S. sclerotiorum* treated with **F15** became abnormally collapsed and shriveled, thereby inhibiting the growth of the hyphae. Furthermore, **F15** exhibited favorable inhibition against the succinate dehydrogenase (SDH) of the *S. sclerotiorum* (IC_50_ = 12.5 μg/mL), and the combination mode and binding ability between compound **F15** and SDH were confirmed by molecular docking. In addition, compound **F11** showed excellent nematicidal activity against *Meloidogyne incognita* at 200 μg/mL, the corrected mortality rate was 93.2%, which is higher than that of tioxazafen.

## 1. Introduction

Plant diseases that are caused by fungi and nematodes have become increasingly serious and have exerted an enormous impact on the viability and health of plants [1,2,3]. To make matters worse, the combination of fungi and nematodes tends to produce a synergistic interaction, and the resultant crop loss from this interaction is greater than the loss caused by each pathogen alone or additive effects [4,5]. Currently, more than 8000 species of fungi are known to cause plant diseases [6], including *Fusarium graminearum*, *Botrytis cinerea* (*B. cinerea*), *Sclerotinia sclerotiorum* (*S. sclerotiorum*), *Rhizoctonia solani*, *Blumeria spp.*, *Pythium spp*., *Colletotrichum spp*., *Fusarium spp*., *Puccinia spp*., *Phytophthora spp*., etc., and these fungi have brought about 85% plant diseases [2] and resulted in large economic crop losses [6]. Meanwhile, *Meloidogyne incognita* (*M. incognita*), *Aphelenchoides besseyi* (*A. besseyi*), and *Bursaphelenchus xylophilus* (*B. xylophilus*) are plant parasitic nematodes that led to serious damage in agricultural crops, cereal crops, and forestry, respectively [1,7]. It has been reported that the annual global agricultural economic losses caused by plant parasitic nematodes is estimated to be USD 157 billion [8]. Therefore, eliminating plant diseases caused by fungi and nematodes has become a global issue in the agricultural sector. Up until now, the most efficient and useful strategy to control fungi and nematodes is chemical prevention. However, the abuse of existing pesticides has led to the emergence of drug-resistant microorganisms, thereby further exacerbating the comprehensive governance of this situation and potentially elevating a huge risk to human health [9,10,11,12]. Moreover, there are few pesticide chemicals that are able to control fungi and nematodes simultaneously. Therefore, developing new, non-resistant pesticides will be of great importance to effectively control these plant diseases

For the past few years, succinate dehydrogenase inhibitors (SDHI) as a key area of new compound research are experiencing fast market growth [13,14]. The SDHI mechanism of action is to inhibit the succinate dehydrogenase (SDH), which plays a crucial role as the only membrane protein in both the tricarboxylic acid cycle and mitochondrial electron transfer chain [15,16]. Among them, fluopyram is a novel fungicide and nematicide with a distinct structure, which was successfully developed by Bayer in 2012; it has an amide bridge can be combined with SDH [17,18]. As a fungicide, it has not been found to be cross-resistant with other fungicides or SDHI fungicides [19,20]. Thus, the modification of the amide-bridged chain is likely to bring out a novel binding mode, further slowing down the development of resistance [14,21]. For all we have seen, the fungicides fluopimomide, florylpicoxamid, and thifluzamide and the nematicides fluazaindolizine and cyclobutrifluram all contain an amide bond (Figure 1). Therefore, it is necessary to introduce amide fragments in the design of new compounds. Fluopyram has become the most promising fungicide and nematocide; however, the application cost of fluopyram is higher than others in agriculture, with the cost of 480 g ha^−1^ being about USD 650 [22].

Heterocyclic compounds have been widely considered in drug design. As one of the most important heterocyclic compounds, 1,2,4-oxadiazole derivatives have exhibited a wide range of biological activities, including herbicidal [23,24], antibacterial [25,26], antifungal [27,28,29], insecticidal [30,31], and other biological activities [32,33,34,35]. Among them, tioxazafen, as the representative nematicide with a 1,2,4-oxdiazole as a core moiety, designed by Monsanto, acts as a new-type seed treatment agent for controlling nematodes [36]. Field trials have shown that it has a good control effect on crop root-knot nematodes, and the 1,2,4-oxadiazole nematicide is considered to have the most market development prospects by the industry at present. In addition, some studies from the literature have also reported that 1,2,4-oxadiazole derivatives have excellent nematicidal activity (Figure 1) [37]. However, little works have been performed on 1,2,4-oxadiazole derivatives against both fungi and nematodes.

In this work, a series of novel 1,2,4-oxadiazole derivatives containing amide fragments with antifungal and nematicidal activities were obtained by introducing amides into 1,2,4-oxadiazole (Figure 2). In addition, the SDH inhibitory ability of the most active compounds and its mechanism of action on fungi were also studied.

## 2. Result and Discussion

### 2.1. Chemistry

The synthetic route of title compounds **F1**–**F24** is shown in Figure 1. First, the different substituted nitriles were reacted with hydroxylamine hydrochloride to obtain the amidoxime a [38]. Then, compound b was obtained from the reaction of intermediate a and ethyl oxalyl monochloride in acetonitrile, and then hydrolysis with LiOH in EtOH was enacted to obtain key intermediate c using methods according to previous methods in the literature [39,40]. Following this, intermediate c was reacted with oxalyl chloride to produce intermediate d [40], which in turn was reacted with the primary amine to obtain compounds **F1**−**F24**.

### 2.2. In Vitro Antifungal Bioassay

The results of the antifungal activity at 50 μg/mL are shown in Table 1. As shown in Table 1, the results demonstrated some compounds with moderate to excellent mycelial growth inhibition activities. In detail, compounds **F3**, **F15**, **F18**, and **F20** showed good antifungal activity against *B*. *cinerea*, with inhibition rates of 56.8%, 58.2%, 55.9%, and 55.8%, respectively, which were lower than those for fluopyram (87.3%) and thifluzamide (80.3%). It is worth mentioning that compounds **F1**, **F3**, **F9**, **F14**, and **F15** exhibited remarkable antifungal activity against *S*. *sclerotiorum*, with inhibition rates of 73.2%, 84.8%, 61.1%, 65.2%, and 89.3%, where the EC_50_ values (Table 2) were 20.8, 5.4, 18.7, 15.3, and 2.9 μg/mL, respectively. Specifically, compound **F15** showed the best antifungal activity against *S*. *sclerotiorum* (Figure 3) and was obviously superior to thifluzamide (4.3 μg/mL) and comparable to fluopyram (1.2 μg/mL).

The results of the structure–activity relationship analysis showed that when the 3-position of 1,2,4-oxadiazole was substituted by an aromatic benzene ring with *n* = 2, the anti-*B*. *cinerea* and anti-*S. sclerotiorum* activities of the compounds with Py substituent in R’ were higher than other compounds with 3-Cl-5-CF_3_-2-Py, such as **F1** > **F2**, **F5** > **F6**, and **F9** > **F10**. However, the activity was opposite when the 3-position of 1,2,4-oxadiazole was heterocyclic and when *n* = 2 (**F14** > **F13**, **F18** > **F17**, **F22** > **F21**). Furthermore, the best activity was found for compounds with *n* = 1 and R′ = 2,4-di-F-C_6_H_3_ when the R group was the same, such as in compounds **F3** > **F1**, **F2**, **F4** or **F15** > **F13**, **F14**, **F16**. In addition, the activity of the compounds with R = thienyl was better than the activity observed in the other groups where R′ = 2,4-di-F-C_6_H_3_, *n* = 1, for instance, **F15** > **F3** > **F7** > **F11** > **F19** > **F23**. Furthermore, when R′ = 2-Py, *n* = 2, electron-donating group or electron-withdrawing group on phenyl both went against the inhibitory to *S. sclerotiorum*, as compounds **F1** (R = C_6_H_5_) > **F9** (R = 4-Cl-C_6_H_4_) > **F5** (R = 4-CH_3_-C_6_H_4_). At the same time, when R′ = 2,4-di-F-C_6_H_3_, *n* = 1, the compound with phenyl in R has more antifungal activity against both *B. cinerea* and *S. sclerotiorum*. In addition, lipophicity affects antifungal activity, such as **F3** (R = C_6_H_5_) > **F7** (4-CH_3_-C_6_H_4_) > **F11** (4-Cl-C_6_H_4_), with the CLogP value (according to Lipinski’s rule, CLogP < 5) are 3.8, 4.3, and 4.5, respectively. These results illustrated that a large steric hindrance of the R group was adverse to fungicidal activity. Meanwhile, when the R was the same, the compounds with R′ = 2,4-di-Cl-C_6_H_3_ and *n* = 0 showed the lowest activity, examples of which can be seen in compounds **F15** > **F14** > **F13** > **F16**.

### 2.3. Cytotoxicity Assays

In order to prove the safety of these compounds to human body, the cytotoxicity of compound to human normal liver L-02 cells was determined by 3-(4,5-dimethylthiazol-2-yl)-2,5-diphenyltetrazoliumbromide (MTT) assay. As shown in Table 2, these compounds showed low cytotoxicity to cells L-02 at 50 and 100 μg/mL, and the cytotoxicity was significantly lower than that of the control thifluzamide, among which compound F9 had the least toxicity. In addition, when the concentration increased to 200 μg/mL, the cytotoxicity of these compounds to cells L-02 enhanced, but all of them were lower than the positive control fluopyram. Especially, the cytotoxicity of highly active compound **F15** to cells L-02 was always lower than that of positive control fluopyram and thifluzamide, which preliminarily indicated that they are low toxic to human liver cells L-02.

### 2.4. In Vivo Anti-S. sclerotiorum Bioassay

Regarding antifungal activity in vitro, compound **F15** showed excellent fungicidal activity against *S. sclerotiorum*. As such, the curative and protective effects of this compound in vivo were evaluated, and the results are listed in Table 3. As shown in Table 3, compound **F15** showed good curative effects at 100, 50, and 25 μg/mL, obtaining the values of 62.3%, 50.0%, and 27.7%, respectively, but these values were still lower than those that were obtained for fluopyram (100 (74.1%), 50 (71.4%), and 25 (65.0%) μg/mL). The protective effects of **F15** at 100, 50, and 25 μg/mL were 71.0%, 66.0%, and 56.3%, respectively, which was close to fluopyram (100 (75.6%), 50 (69.3%), and 25 (66.8%) μg/mL). It is worth noting that the protective and curative effects of compound **F15** showed concentration-dependent properties with these data, and **F15** was demonstrated to be safe for cole leaves at high concentrations (Figure 4).

### 2.5. Nematocidal Bioassay

The nematicidal activity of the novel 1,2,4-oxadiazole derivatives containing amide substructures are illustrated in the Table 4.

As shown in Table 4, some of these target compounds showed significant nematicidal activity against *M. incognita*. Compound **F11** demonstrated excellent nematicidal activity against *M. incognita* at 48 h with a corrected mortality rate of 93.2% at 200 μg/mL, which was higher than that of the positive control tioxazafen (23.9%). At the same time, **F3**, **F6**, **F10**, **F13**, **F14**, and **F20** exhibited good nematicidal activity against *C. elegans* with a mortality rate of 100% being demonstrated at 200 μg/mL. Specifically, **F3** and **F6** showed excellent activity against *C. elegans*, demonstrating a 100% mortality rate at 50 μg/mL, which is better than tioxazafen (90.4%) and similar to the mortality rate of the commercial nematicides fosthiazate and fluopyram. Unfortunately, the target compounds demonstrated a certain amount of nematicidal activity against *A*. *besseyi* and *B*. *xylophilus* at high concentrations.

The structure relationship of the nematicidal activity was observed according to these results. Based on Table 4, when the 3-position of 1,2,4-oxadiazole was substituted by the benzene ring and when R′ = 2,4-di-F-C_6_H_3_, the against *M. incognita* activity of the compounds with R = 4-Cl-C_6_H_4_ was higher than it was for compounds where R = C_6_H_5_ or R = 4-CH_3_-C_6_H_4_, such as in **F11** > **F3** > **F7**. At the same time, the compounds with *n* = 1 or *n* = 0 had higher against *B. xylophilus* activity than the compounds with *n* = 2 when R and R′ were the same, such as in **F3**, **F4** > **F1**, **F2** or **F23**, **F24** > **F21**, **F22**. In addition, it was found that when R or R′ were the same, the against *M. incognita*, against *B*. *xylophilus,* and against *C. elegans* activity of *n* = 1 was higher than that for the compounds where *n* = 0 (except R = 6-Br-3-Py), for example, in **F3** > **F4**, **F7** > **F8**, **F11** > **F12**, **F15** > **F16**, **F23** > **F24**. However, the activity against *A*. *besseyi* showed that the structure–activity relationship was not significant. Generally speaking, the structural activity relationship among different species of nematodes was not consistent.

### 2.6. Scanning Electron Microscopy (SEM) of Compound ***F15*** on the Hyphae Morphology

The mycelial morphology of *S. sclerotiorum* that was determined using compound **F15** and 0.1% DMF was observed using SEM (Figure 5). The negative control group showed a typical and characteristic morphology, with uniform and linear hyphae relative to obvious collapse. However, the hyphae of *S. sclerotiorum* became abnormally collapsed and shriveled after treatment with 50 μg/mL **F15**. This indicates that compound **F15** may affect the morphology of mycelium by destroying the cell membrane or cell wall of *S. sclerotiorum*, affecting the further growth and reproduction of *S. sclerotiorum*.

### 2.7. Enzymatic Inhibition Activity of SDH

Compound **F15** had the best anti-*S. sclerotiorum* activity, and its structure was shown to similar to that of the SDH inhibitor fluopyram. To further illustrate how compound **F15** is a potential SDH inhibitor, the SDH inhibitory activity of compound **F15** was tested, and fluopyram was chosen as the positive control. As shown in Table 5, **F15** displayed good SDH inhibitoty activity, with and IC_50_ value of 12.5 μg/mL, which was close to that of fluopyram (7.9 μg/mL). The result provided a basis for the design of compounds as potential SDH inhibitors.

### 2.8. Molecular Docking Study

To explain the binding mode of compound **F15** to the SDH receptor, a molecular docking study on compound **F15** was conducted, and fluopyram was selected as the comparative standard. As seen in Figure 6A,B, compound **F15** and fluopyram are both able to connect to the surrounding amino acid residues of the active pocket well via conventional hydrogen bonds and π–π interactions, and they demonstrate similar conformations in the active protein pocket on the SDH. Both compounds have strong hydrogen bonds actions with the residues O/Trp-173 and Q/Tyr-58, which are extremely important for the stability of the combination of SDH inhibitors and SDH. The hydrogen bond distance formed by **F15** (2.7 and 2.8 Å) with the residues O/Trp-173 and Q/Tyr-58 is shorter than fluopyram (2.8 and 2.9 Å), which indicates that **F15** has stronger hydrogen bond interactions with these two residues. Furthermore, the indole ring of residue O/Trp-173 forms a π–π stack interaction with the aromatic ring of **F15**. The binding energy ΔG_bind_ between compound **F15** and SDH was −86.8 kcal/mol (Table 6), which was similar to fluopyram (−88.1 kcal/mol), which demonstrates that the introduction of a hydrophobic group on the oxadiazole ring will help the compound well with bind to the protein.

The root mean square deviation (RMSD) value, which is a guiding factor for the structural stability of the complex structure and its movement in the simulation process [41,42], was calculated by molecular dynamic simulation (MDS), and the results are shown in Figure 7. At the beginning, it was found from the RMSD that **F15** was more stable than that of fluopyram at 1 ns, which was caused by the 180 rotation of fluopyram’s benzene ring (Appendix A), with the ΔE_vdw_ of −143.7 kcal/mol, which was lower than that of **F15** (−137.1 kcal/mol). Therefore, it is speculated that this difference is influenced by Van der Waals forces. In addition, the RMSD curves of fluopyram and **F15** tend to be close after 8.3 ns, indicating that the ligand movement in the protein is becoming more stable. Surprisingly, compound **F15** has three large fluctuations at 3.2, 8.2, and 9.1 ns. The first violent fluctuation can be observed at 3.2 ns and was due to the break in the hydrogen bond between the nitrogen at the 2-position of 1,2,4-oxadiazole and the hydroxyl group on the benzene ring of the residue Tyr-58, which broke and then formed a hydrogen bond with Ser-170; at the same time, after the hydrogen bond between the carbonyl group on the amide bond and Trp-173 broke, and it formed a new hydrogen bond with Pro-169, leading to the distortion of the entire small molecule (Appendix A). The severe fluctuation at 8.2 ns was caused by the hydrogen bond formed by oxygen at the 1-position of 1,2,4-oxadiazole and the breaking of Arg-43 (Appendix A). The fluctuation at 9.1 ns was different from the previous two fluctuations due to the position shift of the thiazole ring, so the RMSD fluctuation was smaller than the first two fluctuations (Appendix A). Therefore, hydrogen bonding plays an important role in the stable combination of small molecules and proteins. Therefore, hydrogen bonding plays an important role in the stable combination of small molecules and proteins. The results indicate that the binding mode of **F15** and SDH is similar to that of the SDH inhibitor fluopyram and confirms that compound **F15** has excellent anti-*S*. *sclerotiorum* activity.

## 3. Materials and Methods

### 3.1. Chemistry

#### 3.1.1. Instruments and Chemicals

All solvents and chemical reagents were purchased from Aladdin Reagent (Shanghai, China) and Energy Chemical (Shanghai, China), respectively. The melting points of all novel compounds were determined on an X-4B microscope melting point apparatus and were uncorrected (Shanghai Electrophysics Optical Instrument Co., LTD, Shanghai, China). Reactions were detected by thin-layer chromatography (TLC) and visualized under UV light at 254 nm. All ^1^H NMR and ^13^C NMR spectra data were recorded on a Bruker DPX-500 or a DPX-400 spectrometer (Bruker, Billerica, MA, USA), DMSO-*d_6_* or CDCl_3_ were used as solvents, and tetramethylsilane was used as an internal standard. The high-resolution mass spectrometer (HRMS) data of the compounds were obtained using a Thermo Scientific Q-Exactive (Thermo Scientific, Missouri, MO, USA). The purity of that compounds was detected by HPLC, which was performed on a Shimadzu LC-2030 Plus (Shimadzu, Tokyo, Japan) with a Daicel Chiralpak AD-H column (conditions: 10% Isopropyl alcohol/Hexanes, 1.0 mL/min, λ = 220 nm, 30 °C). Chromatography was conducted on silica gel 200–300 mesh (Fluka, Daicel (China) investment Co., LTD, Shanghai, China) and under low pressure. The general experimental procedures that were used for the synthesis of all of the compounds is described in the following paragraphs.

#### 3.1.2. General Procedure for the Synthesis of Intermediate **a1**–**a6**

In a 100 mL three-necked bottle, hydroxylamine hydrochloride (33.0 mmol) from commercial sources was added to EtOH (30 mL). Sodium hydroxide solution (*v/v* = 1/2) was then dropped into the reaction mixture at room temperature. Then, aromatic nitrile or heterocyclic nitrile (30.0 mmol) was dissolved in ethanol (20 mL) and was then added to the flask. The mixture was heated at 75 °C for 5–7 h and was monitored by means of thin-layer chromatography (TLC). The precipitated product was filtered off by suction, the filtrate was concentrated under reduced pressure, and the residue was then diluted with ethyl acetate, washed with water (20 mL × 2), saturated with brine (20 mL × 2), and extracted with ethyl acetate (20 mL × 2). The organic layer was dried over anhydrous sodium sulfate. The solution was concentrated to produce intermediate **a1**–**a6**.

#### 3.1.3. General Procedure for the Synthesis of Intermediate **b1**–**b6**

Compound a (22.0 mmol) and triethylamine (33.0 mmol) were added to 20 mL of acetonitrile by means of stirring, and then the mixture was cooled to 0–5 °C. This temperature was maintained, and ethyl chlorooxalate (26.0 mmol) was added dropwise into the above mixture. Then, the reaction mixture was stirred at 0–5 °C for 0.5 h, and it was then refluxed for 5–8 h and was monitored by means of TLC. The solvent was removed under vacuum conditions, and the residue was diluted with dichloromethane (DCM) and washed with brine (150 mL × 2), the organic phase was separated and dried, and it was further purified to create the white solid of **b1**–**b6** by column chromatography (ethyl acetate/petroleum ether = 12/1).

#### 3.1.4. General Procedure for the Synthesis of Intermediate **d1**–**d6**

An amount of 2 N LiOH was added to a solution of carboxylates b (14.0 mmol) in ethanol at room temperature, and this solution was stirred for 0.5–1 h. Additionally, the ethanol was removed in vacuo, 1 N HCl was used to adjust the pH = 5–6, the white solid precipitated, and it was then collected by filtration, which was dried to obtain the carboxylic acid c1–c6 and proceed directly to the next step without purification. Intermediate **c** (2.0 mmol) was dissolved in dry 2 mL DCM in an ice bath, (COCl)_2_ (4.0 mmol) was slowly added, and then one drop of *N*, *N′*-dimethylformamide (DMF) was dropped into the mixture. Then, the mixture was removed from the ice bath and was stirred at ambient temperature for 6 h. The resulting mixture was concentrated in vacuo to produce intermediate **d1**–**d6**.

#### 3.1.5. General Procedure for the Synthesis of Target Compounds **F1**–**F24**

Primary amine (1.0 mmol), Et_3_N (1.5 mmol), and DCM (2 mL) were added into a round-bottom flask, and chloride d was dissolved in DCM (1 mL) and dropped into the above reaction mixture. The DCM from the reaction mixture was removed in vacuo after stirring at 25 °C for an additional 8–12 h. The residue was dissolved in ethyl acetate (30 mL), and the organic layer was then washed with water and saturated with sodium bicarbonate and brine successively. Anhydrous Na_2_SO_4_ was used to dry the organic layer, and it was then filtered. The solution was concentrated under vacuum conditions, and the residue was purified by column chromatography (ethyl acetate/petroleum ether = 20/1) to obtain the target compounds (**F1**–**F24**) with yields of 43.3–67.3%.

### 3.2. Antifungal Activity Bioassay In Vitro

*B*. *cinerea* and *S. scleotiorum* were kindly provided by the Department of Pesticides, College of Plant Protection, Nanjing Agricultural University (Nanjing, China). All compounds were evaluated for mycelium growth inhibition tests against the two plant pathogenic fungi mentioned in previous reports [43,44]. These two fungi were inoculated on potato dextrose agar (PDA) plates and were grown in biochemical incubators at 25 ± 1 °C for 2–4 days. The new mycelium was used to determine the antifungal activity. The target compounds were dissolved in 100 µL DMF, and they were then added into the PDA media to make the concentration 50 µg/mL. Mycelia dishes of about 5 mm in diameter were cut from the culture medium. Mycelium were picked up with a sterile inoculation needle, and the sample was inoculated in the middle of a PDA plate in a sterile environment. Afterwards, a preliminary screening of fungal activity was performed. The commercial fungicides thifluzamide and fluopyram were chosen as positive controls, and the negative control group was prepared without compounds using the same methodology. Each test was carried out three times. The radial growth of the fungal colonies was gauged and recorded when the diameters of the blank control mycelia reached 5.0–5.5 cm. The compounds with superior activity were further evaluated through the determination of the EC_50_ value. The final data were analyzed using SPSS v.25.0 (IBM, New York, NY, USA). The following formula was used to calculate the fungi inhibition rate.
Inhibition rate (%) = [(Mycelium diameter of negative control − Mycelium diameter of treatment)/(Mycelium diameter of negative control − 0.5)] × 100%(1)

### 3.3. Cytotoxicity Assays In Vitro

The human liver L-02 cells were obtained from the biology laboratory in Key Laboratory of Chemistry for Natural Products of Guizhou Province and Chinese Academy of Sciences (Guiyang, China). The cells were grown in Dulbecco’s Modified Eagle’s Medium (supplemented with 10% (*v/v*) FBS, 100 U/mL penicillin, and 100 μg/mL streptomycin) at 37 °C in a CO_2_ incubator (5% CO_2_ and 95% air, 95% humidity). The effect of some compounds on cell viability was determined with the MTT assay according to the previously published literature [45,46]. A total of 5000 cells were inoculated in a 96-well plate and three wells of each concentration of test compound were used. A total of 24 h after inoculation, the cells had grown to ~60% confluency, and different concentrations of compounds were added, and then the cells were cultured for 48 h. DMSO was used as the control. A total of 20 μL of MTT solution was added to each well and incubated at 37 °C for 4 h. Formazan crystals were dissolved in 150 μL of DMSO and quantified using a microplate reader (Thermo Scientific, Vario Skan Flash, Shanghai Bio-Gene Technology Co., LTD, Shanghai, China) at 490 nm. The cytotoxicity was calculated according to following formula:Cytotoxicity (%) = [(O_CK_ − O_T_)/O_CK_] × 100%(2)
where O_CK_ represents the optical density (OD) value of the blank control group, and O_T_ represents the OD value of experimental group.

### 3.4. Nematicidal Activity Bioassay In Vitro

*M*. *incognita* (parasitic on tomato plants), *A*. *besseyi* (potato dextrose agar–*Botrytis cinerea* breeding), *B*. *xylophilus* (potato dextrose agar–*Botrytis cinerea* breeding), and *C*. *elegans* (nematode growth medium–inactivated *Escherichia coli* OP_50_ cultivation) were provided by the Fine Chemical Research and Development Center of Guizhou University (Guizhou, China). The methodology used for the nematocidal bioassays of these target compounds was modified in the light of the conventional methods reported in the previously published literature [47,48,49]. The nematicidal activity of the target compounds against four nematodes was determined using 48-well biochemical culture dishes. All of the compounds were dissolved with 100 μL DMF and were diluted with 1% Tween-80 to obtain 200 and 50 μg/mL concentrations for the bioassays. Fosthiazate and tioxazafen were used as positive controls at the same concentrations, and a negative control group was prepared without compounds using the same methodology. An amount of 10 µL of nematode suspension (about 50 nematodes) and 300 µL of the trial solution were added to corresponding hole, and each treatment was set to three repetitions. All of the 48-well plates were placed in a biochemical incubator at 27 °C for dark light culture. After 48 h, the dead nematodes were counted under a stereomicroscope, and the mortality was calculated (if the nematode did not move when it was touched with a needle, it was considered dead).
Corrected mortality (%) = [(mortality of treatment % − mortality of negative control %)/(1 − mortality of negative control %)] × 100%(3)

### 3.5. Antifungal Activity Bioassay In Vivo

Susceptible cole leaves collected from Guizhou Academy of Agricultural Sciences were used to determine the in vivo efficacy of **F15** according to previous literature [50,51]. Leaves that were of uniform size and similar shape were resected from healthy rapeseed plants, which were sterilized with 1% sodium hypochlorite for 1 min, and they were then rinsed with sterilized water and were blotted with sterile filter paper. Vaccination time was used to distinguish between the protective activity assays and curative activity assays. A 5 mm mycelia dish was placed at the widest center of the cole leaf, and it was ensured that it was not placed on the main vein of the leaf. For the protective activity assay, test compounds with different concentrations were sprayed on the leaves until the liquid reached the surface. Inoculation was conducted after 24 h. For the curative activity assay, which took place 24 h after inoculation, compounds of different concentrations were sprayed on the surfaces of the leaves until the liquid flowed. Fluopyram and sterile distilled water were used as a positive control and negative control, respectively. All of the inoculated leaves were placed inside of a 25 °C light incubator with 85% relative humidity and a 16 h photoperiod for disease development. The average diameters of the lesions were measured using the cross method after 72 h. The disease control efficacy was obtained through the following formula:Control efficacy (%) = (1 − D_T_/D_CK_) × 100%(4)
where D_CK_ represents the lesion diameter of the sterile distilled water control, and D_T_ represents the lesion diameters of the treatment samples. Each test was carried out five times, and the experiment was performed three times.

### 3.6. SDH Enzyme Activities Bioassay

The succinate dehydrogenase assay kit (Comin, Jiangsu, China) was used to determine the enzyme activity of **F15**, which was assessed as reported previously [21,52,53]. *S*. *sclerotiorum* was grown in potato dextrose (PD) medium containing **F15** with different concentrations to test the IC_50_ value of SDH. Fluopyram was used as a positive control. All operations were strictly in accordance with the operating instructions of the SDH assay kit. The absorbance of each treatment on fungal SDH was measured at 600 nm to determine the inhibitory effects of each treatment on fungal SDH.

### 3.7. SEM Observations

SEM observations of the hyphae of *S. sclerotiorum* were implemented in line with reported methods [51,54]. Mycelium disks with a diameter of 5 mm were taken from the edge of PDA medium containing 50 μg/mL of **F15** and were incubated for 2 days at 25 °C, and PDA with 0.1% MDF was used as a control. To fix the samples, 2.5% glutaraldehyde was used, and samples were fixed at 4 °C for 1 day and were rinsed for 15 min with 0.1 M phosphate buffer, a process that needed to be repeated three times, and the samples were then fixed with 1% OsO_4_ solution for 1 h and then dehydrated in 10%, 30%, 50%, 70%, 90%, and 100% ethanol for 10 min each time. Finally, gold coating was carried out after drying at the critical point, and then the samples were observed under a scanning electron microscope (Nova Nano SEM 450, Hillbora, OR, USA).

### 3.8. Molecular Docking

A molecular docking station was built using the Ledock program according to the literature [14,51]. At present, due to the lack of reports on the tri-dimensional structures of the SDH of the fungal species, the crystal structure of SDH (PDB ID: 2FBW) [14,21,48] was downloaded from *Gallus gallus* on the Protein Data Bank (https://www.rcsb.org, las accessed on 18 October 2021) and was processed with Pymol. The molecular structures of **F15** and fluopyram were drawn using ChemBioDraw Ultra 14.0 software and were optimized to minimize energy. A 17.5 × 15.3 × 14.7 docking box was generated with the Carboxin Standard in the protein as the center, and the docking station generated 20 ligand conformations with an RMSD less than 1.0 Å. The docking results were visualized in 3D by the Pymol software v.2.4.0 [55]. In addition, the binding free energies were calculated by making use of the molecular mechanics Possion–Boltzmann surface area (MM/PBSA) method [56,57].

To vividly explain the interaction mechanism and the binding stability between compound **F15** or fluopyram and SDH, MDS was performed to revise the docking results, and Gromacs software (version 2020.5) was used for the MD simulations [41,51,58]. The force fields of these two compounds were obtained at https://www.bio2byte.be/research/, last accessed on 18 October 2021), and water was then added under the stand of the compound “amber96sb.ff”. Energy minimization was executed by MD simulation when the electrons were in equilibrium, the temperature was raised to 300 K, the pressure was increased to one atmosphere, and finally, the MD simulations were performed for 10 ns.

## 4. Conclusions

In this work, 24 1,2,4-oxadiazole derivatives containing an amide substructure were designed and synthesized, and then, their fungicidal and nematicidal activity were determined. The antifungal results revealed that compound **F15** displayed the highest in vitro antifungal activity against *S*. *sclerotiorum,* with an EC_50_ value of 2.9 μg/mL. In vivo test indicated that compound **F15** could control the disease caused by *S. sclerotiorum* that infected cole leaves, demonstrating curative effect and protective effects of 62.3% and 71.0% at 100 μg/mL. Preliminary studies on its anti-fungal mechanism have shown that compound **F15** could cause the obvious collapse and shrinkage of the hyphal morphology of *S. sclerotiorum*. The SDH inhibitory activity and the molecular docking results co-indicated that compound **F15** is a potential SDH inhibitor. Furthermore, the in vitro nematocidal bioassays indicated that some compounds showed significant nematicidal activity against the four nematodes discussed above. As such, the 1,2,4-oxadiazole framework can be considered a comfortable model that can be used to find highly efficient alternative candidates against plant diseases that are caused by fungi and nematodes.

## Data Availability

All data generated in this study is presented in the current manuscript. No new datasets were generated. Data are available upon request from the corresponding author.

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
