# Peer review of "Design, Synthesis and Antifungal/Nematicidal Activity of Novel 1,2,4-Oxadiazole Derivatives Containing Amide Fragments"

_ijms, 2022, doi:10.3390/ijms23031596_

Round 1

Reviewer 1 Report

The Manuscript “Design, Synthesis and Bioactivity Evaluation of Novel 1,2,4-Oxadiazole Derivatives Containing Amide Fragment” propose some 1,2,4-oxadiazoles with amide functionality as solution to the joint control of fungi and nematodes. The results revealed that compound F15 displayed the highest in vitro antifungal activity against S. sclerotiorum. Moreover, preliminary studies have shown that compound F15 could cause collapse and shrinkage of the hyphal morphology of S. sclerotiorum by SDH inhibition. Furthermore, in vitro nematocidal bioassays indicated some compounds had significant nematicidal activities.

On the whole the topic is interesting and the research is well conducted, some major points I would raise are the following:

  1. Page 3 the authors state :“which showed that groups with large steric hindrance are not conducive to antifungal activity”, I wonder if you can  envisage any further electronic/lipophilic effect in order to justify such behavior.
  2. Page 2 notes [ 23-25] is poorly representative of the enormous literature on this topic, please add reviews or book chapter concerning bioactive 1,2,4-oxadiazoles.
  3. Considering these compounds are proposed to protect crops, for human use, it would be opportune to conduct toxicity test in vitro.

Besides these major points some minor English style suggestions are the following:

Page 2 “. For all we seen, such as the fungicides” change as follows “. For all we have seen, such as the fungicides”

Page 2 “Heterocyclic compounds have been widely concerned in drug design Heterocyclic compounds have been widely concerned in drug design always” change as follows “Heterocyclic compounds have always been widely considered in drug design”

Page 2 “Among them, tioxazafen as the representative nematicide with a 1,2,4-oxdiazole as a core moiety was designed by Monsanto, which acts as a new-type seed treatment agent that control of nematodes [26].” change as follows: “Among them, tioxazafen, as the representative nematicide with a 1,2,4-oxdiazole as a core moiety, designed by Monsanto, which acts as a new-type seed treatment agent for nematodes control [26].”

Page 2 “However, little works have been performed on 1,2,4-oxadiazole derivatives joint against fungi and nematodes.” change as follows “However, little works have been performed on 1,2,4-oxadiazole derivatives against both fungi and nematodes.”

Page 2 “To search for a novel and low-cost compounds with good activity for resist fungi and nematodes. The amide moiety was introduced into 1,2,4-oxadiazole to obtain a series of novel 1,2,4-oxadiazole derivatives containing amide fragment (Figure 2).” change as follows” In order to search for novel and low-cost compounds with good activity against fungi and nematodes, the amide moiety was introduced into 1,2,4-oxadiazole and  a series of novel 1,2,4-oxadiazole derivatives containing amide fragment have been obtained (Figure 2).”

Page2-3 “And all the title compounds have further evaluated their antifungal and nematicidal activities.” change as follows “All the title compounds have been further evaluated for their antifungal and nematicidal activities.”

Page 3 “The synthetic route of title compounds F1−F24 are shown in Scheme 1” change as follows “is shown”

Page 3 “Then, compound b was got from intermediate a and ethyl chlorooxalate in acetonitrile, and then hydrolysis with LiOH in EtOH to obtain key intermediate c according to the literature [29,30].” Please change as follows “Then, compound b was obtained from intermediate a and ethyl chlorooxalate in acetonitrile, and then hydrolysis with LiOH in EtOH afforded key intermediate c according to the literature [29,30].”

Page 3 “Followed, intermediate c was reacted with oxalyl chloride to give intermediate d, which was reacted with primary amine to obtain compounds F1−F24, and their structures were confirmed by 1H NMR, 13C NMR, and HRMS data and listed in the Supporting Information.”  Change as follows” Following, intermediate c was reacted with oxalyl chloride to give intermediate d, which in turn was reacted with primary amine to obtain compounds F1−F24; all the products structures were confirmed by 1H NMR, 13C NMR, and HRMS data as listed in the Supporting Information.”

Page 3 “but which lower than that of fluopyram (87.3%)” delete “which”

Page 3 “which EC50 values (Table 2 and Figure 3) were” change as follows” whose EC50 values (Table 2 and Figure 3) were”

Page 4 “ R' = 2,4-diF-Ph” In my opinion the abbreviation diF is not correct

Page 4 “these results illustrated that large steric hindrance of the R group unfavor to fungicidal activity along.” change as follows “these results illustrated that large steric hindrance of the R group unfavors fungicidal activity.”

Page 4 “, for example these compounds, F15 > F14 > F13 > F16” change as follows “, as compounds F15 > F14 > F13 > F16”

Page 7 “However, the hyphae of S. sclerotiorum went into abnormally collapsed and shriveled treatment with 50 μg/mL F15.” change as follows “However, the hyphae of S. sclerotiorum became abnormally collapsed and shriveled upon treatment with 50 μg/mL F15.”

Page 8 figure caption “Figure 5. Scanning electron micrographs of S. sclerotiorum hyphae in untreated control and 50 μg/mL compounds F15 treated.” change as follows “Figure 5. Scanning electron micrographs of S. sclerotiorum hyphae in untreated control and treated with compound F15 at 50 μg/mL.”

Page 8 “The result provided a foundation for designed compounds as potential SDH inhibitors.” change as follows “The result provided a basis  for the design of compounds as potential SDH inhibitors.”

Author Response

Dear Review,

Thank you very much for your attention throughout the evaluation process of our manuscript “Design, Synthesis and Bioactivity Evaluation of Novel 1,2,4-Oxadiazole Derivatives Containing Amide Fragment” submitted to International Journal of Molecular Sciences. The manuscript has been carefully modified following your comments. Each comment has been addressed and answered point by point in the revised manuscript. These revisions are colored red in the revised manuscript and also listed below for your reference. The content was listed in attachment.

Reviewer 2 Report

The presented manuscript describes the design and evaluation of a number of amide-containing oxadiazoles as antifungal and nematocidal for plants.

The presented work is authentic, and the study is scientifically sound and well designed. The conclusion is supported with the results and the discussion is appropriate. The conducted experiments in vitro, in vivo and the enzymatic inhibitory activity were sufficient at this stage to prove the antifungal activity.

Few modifications can improve the manuscript:

  1. The title of the manuscript doesn’t reflect the type of biological activity under study. Please add the antifungal and nematocidal activity to the title.
  2. Although the presented work was aimed to explore the antifungal and nematocidal activity of oxadiazole derivatives with main aim to present compounds that can have the dual activity, the study focused on the antifungal activity with very limited presentation of the nematocidal activity. The authors needed to expand their study to explore a structure activity relationship for the nematocidal activity and the IC50 of the best compounds.
  3. The conclusion that some of these compounds have nematocidal activity is not well supported in the results. The designed series had very weak activity. A well thought of SAR could present some ideas for structures that may have better activity.
  4. The introduction needs English language revision.

Author Response

Dear Review,

Thank you very much for your attention throughout the evaluation process of our manuscript “Design, Synthesis and Bioactivity Evaluation of Novel 1,2,4-Oxadiazole Derivatives Containing Amide Fragment” submitted to International Journal of Molecular Sciences. The manuscript has been carefully modified following your comments. Each comment has been addressed and answered point by point in the revised manuscript. These revisions are colored red in the revised manuscript and also listed below for your reference. The contents was listed in attachment.

Reviewer 3 Report

The manuscript “Design, Synthesis and Bioactivity Evaluation of Novel 1,2,4-Oxadiazole Derivatives Containing Amide Fragment” describes the design and synthesis of 24 oxadiazole derivatives containing an amide fragment tested for their antifungal and activities. Among them, compound F15 emerged the most active in inhibiting Sclerotinia sclerotiorum in vitro and in infected cole while a few different compounds showed nematicidal activity, even if at high doses.

Although the topic is interesting, the compounds reported in this work represent a preliminar starting point for a further development of most potent and interesting derivatives. Unfortunately, the manuscript is not well-written and in some parts it is hard to understand the meaning of single sentences. It strongly needs English revision. Thus, it may be considered for publication after major revision.

  • Abstract: line 7, sentence “The bioassays revealed that some target compounds with excellent antifungal activity against Sclerotinia sclerotiorum in vitro”. The sentence written in this form seems not concluded. In case remove “that”
  • Abstract: line 11, sentence “F15 exhibited favorable inhibitory to ..” favourable inhibition or favourable inhibitory potency?
  • Introduction: sentence “…developed by Bayer in 2012, it has an amide bridge can be combined with SDH [17,18] and has not found cross-resistance between another fungicides or SDHI fungicides [19,20].” Rephrase, no sense
  • Introduction: Page 3, sentence “And all compounds have further evaluated their antifungal and nematicidal activities.” Please rephase. No sense
  • Figure 1: There are several errors. 1. In the amide linkage of florylpicoxamid cannot be present a chiral bond; 2. please write all the name of compounds in only one line (for example thifluzamide and cyclobutrifluram must be corrected); 3. to be consistent among all structures presented in the figure, in compound thifluzamide the trifluoromethyl group should be shortened as in compound fluopyram as CF3; 4. the last two compounds in figure must have a number, authors can decide if give them number 1 and 2 with the respective references in apex or the number with which they have been numbered in the reference paper.
  • Paragraph 2.2: “It is worth mentioning that compounds F1, F3, F9, F14 and F15 exhibited a remarkable antifungal activity to S. sclerotiorum with the inhibition rate of 73.2%, 84.8%, 61.1%, 65.2%, and 89.3%, which EC50 values (Table 2 and Figure 3) were 20.75, 5.41, 18.66, 15.27, and 2.89 μg/mL, respectively.” Figure 3 cannot be cited in this point since it refers only to compound 15. Moreover, it is not commented anywhere.
  • Title 2.3 “In Vivo Fungicidal Activities Resist S. sclerotiorum” please rephrase. No sense.
  • Table 4 and Table 5, can they be combined in a unique table? In order to better appreciate and compare compounds activity profile.
  • Supporting Information: Several final compounds among F1-F24 presented fluorine atoms in their structures; why in the described C13 NMR spectra for none compounds the multiplicity and the carbon-fluorine coupling constants have been reported? I suggest authors to review carefully the spectra interpretation.
  • Supporting: Although the final compounds have been fully characterized, for intermediates a-d only a general procedure has been described. I suggest authors to include in the Supporting Information a short description on yields and at least a H NMR of all not already reported intermediates in order to permit others to reproduce the synthesis of final compounds.
  • What about the purity of the synthesized compounds reported in the study? How was it determined? At least a sentence should be given.

Minor comments:

  • Abstract: explain SDH abbreviation
  • Abstract should be maximum 200 words in length, this is 217.
  • Introduction: Pag 2 “To search for a novel and low-cost compounds with good activity for resist fungi and nematodes. The amide moiety was introduced into 1,2,4-oxadiazole to obtain a series of novel 1,2,4-oxadiazole derivatives containing amide fragment (Figure 2).” Remove the point and make a unique sentence.
  • Figure 2: please shift n=0,1,2 under the generic structure F1-F24. No sense to be above the arrow.
  • Paragraph 2.1: Correct the sentence “The synthetic route of title compounds F1−F24 are shown in Scheme 1” with The synthetic routes of title compounds F1−F24 are shown in Scheme 1 or The synthetic route of title compounds F1−F24 is shown in Scheme 1
  • Paragraph 2.2: “the results demonstrated some compounds with moderate to outstanding activities” No compounds showed outstanding activity.
  • Paragraph 2.3 “Among the antifungal activity in vitro” not among, maybe concerning, regarding??
  • Graphical Abstract was not present in the submitted file.

Author Response

(The authors gave the same response as above.)

Round 2

Reviewer 1 Report

The manuscript ijms-1538108 has been
sufficiently improved to warrant publication in IJMS.

Author Response

Dear Review,

Thank you very much for your attention throughout the evaluation process of our revised manuscript “Design, Synthesis and Antifungal/Nematicidal Activity of Novel 1,2,4-Oxadiazole Derivatives Containing Amide Fragments” submitted to International Journal of Molecular Sciences. According to your comments, the English language and style of the manuscript has been revised, these revisions are colored red in the revised manuscript and also listed in attachment. We believe that the revised manuscript is more suitable for publication on International Journal of Molecular Sciences than before.

Reviewer 3 Report

I appreciated all the work done by authors to revise their manuscript following reviewers comments and make it more suitable for publication in International Journal of Molecular Sciences.

I would suggest only a few further corrections to do in the Supplementary Material.

  • Page 1-2, I suggest to remove paragraphs 1.1, and the general procedures 1.2, 1.3, 1.4, and 1.5 since they are already reported in the main manuscript. Instead, I would suggest to leave only the characterization data of not known intermediated as well as target compounds.

Author Response

Dear Review,

Thank you very much for your attention throughout the evaluation process of our revised manuscript “Design, Synthesis and Antifungal/Nematicidal Activity of Novel 1,2,4-Oxadiazole Derivatives Containing Amide Fragments” submitted to International Journal of Molecular Sciences. According to your comments, In the Supplementary Material, “Page 1−2, the paragraphs 1.1, and the general procedures 1.2, 1.3, 1.4, and 1.5 were deleted. Then, “1. Synthesis of the target compound” was changed to “1. The characterization data of the not known intermediated”. In addition “The characterization data of the not known intermediated including 1H NMR and 13C NMR were shown as below” was added after 2 the new paragraph 1. We believe that the revised manuscript is more suitable for publication on International Journal of Molecular Sciences than before. Thank you for your suggestions again.